# Stainless Steel as A Bi-Functional Electrocatalyst—A Top-Down Approach

**DOI:** 10.3390/ma12132128

**Published:** 2019-07-02

**Authors:** Joakim Ekspong, Thomas Wågberg

**Affiliations:** Department of Physics, Umeå University, 90187 Umeå, Sweden

**Keywords:** water splitting, electrolysis, bifunctional, electrocatalysts, hydrogen evolution reaction, oxygen evolution reaction, sustainable, stainless steel, nano

## Abstract

For a hydrogen economy to be viable, clean and economical hydrogen production methods are vital. Electrolysis of water is a promising hydrogen production technique with zero emissions, but suffer from relatively high production costs. In order to make electrolysis of water sustainable, abundant, and efficient materials has to replace expensive and scarce noble metals as electrocatalysts in the reaction cells. Herein, we study activated stainless steel as a bi-functional electrocatalyst for the full water splitting reaction by taking advantage of nickel and iron suppressed within the bulk. The final electrocatalyst consists of a stainless steel mesh with a modified surface of layered NiFe nanosheets. By using a top down approach, the nanosheets stay well anchored to the surface and maintain an excellent electrical connection to the bulk structure. At ambient temperature, the activated stainless steel electrodes produce 10 mA/cm^2^ at a cell voltage of 1.78 V and display an onset for water splitting at 1.68 V in 1M KOH, which is close to benchmarking nanosized catalysts. Furthermore, we use a scalable activation method using no externally added electrocatalyst, which could be a practical and cheap alternative to traditionally catalyst-coated electrodes.

## 1. Introduction

Hydrogen has the highest gravimetric energy density of all known substances [1], where the chemically stored energy can be extracted as electricity in fuel cells with zero CO_2_ emission and water as the only output [2]. However, for the full energy cycle of hydrogen to be sustainable and economical it is essential that the production of hydrogen is efficient and can be realized from non-fossil sources. Hydrogen production by electrolysis of water is a promising zero emission technique, but suffers from relatively low efficiency and the use of scarce noble metals in the process [2]. In this technique, water is split into hydrogen and oxygen by two half-cell reactions, i.e., the cathodic hydrogen evolution reaction (HER) and the anodic oxygen evolution reaction (OER). To increase the water splitting efficiency, electrocatalysts are used to accelerate the involved reactions. Commercial electrocatalysts are today made of scarce noble-metals such as iridium or ruthenium for the OER, and for the HER platinum-based electrocatalysts are the most efficient [2,3,4]. Therefore, for this technique to be cost-effective, finding earth-abundant materials that catalyze the water splitting reactions with high-efficiency is crucial, especially for the OER that suffer from the highest overpotential of the two reactions.

In recent years, abundant transition metals such as nickel and iron have shown promising results for catalyzing the OER in alkaline medium with even lower overpotentials compared to noble-metals, thus increasing the efficiency [3,4,5]. On the HER side, there are many highly efficient abundant electrocatalysts in both alkaline and acidic media, such as transition metals phosphides, nickel and molybdenum-based compounds that could possibly replace platinum in the near future [3,6,7,8,9]. Additionally, in all electrochemical cells, electrons has to be transferred to and from the electrocatalysts for driving the reactions, and in alkaline electrolyzers current collectors of alkali resistant materials such as titanium, stainless steel, or nickel has to be used [10,11]. Typically, electrocatalyst are prepared separately and adhered to these current collectors trying to maintain a good physical connection to keep the contact resistance at the minimum at the interface [11]. One should also strive to reduce the cost of metals and methods used for the current collectors.

In this paper, we aim to improve the activity of ordinary stainless steel as an electrocatalyst for the water splitting reaction in alkaline medium using a top-down approach, without adding any separate electrocatalyst. We must mention that stainless steel recently have shown to be active as an electrocatalyst and in this study we focus to create a nanostructured surface to even improve the high activities measured [9,12,13]. To realize this, we use the fact that stainless steel contains several catalytically active elements such as iron, nickel and molybdenum in the bulk. The outermost layer of stainless steel is however made of insulating chromium, which make it a poor catalyst. To overcome this issue, H Schäfer et al. previously showed that it is possible to migrate the nickel and iron particles to the surface by harsh anodization [12]. Remarkably, their activated stainless steel performed as an OER catalyst with benchmarking performance. To our knowledge, stainless steel has not shown high efficiency as an electrocatalyst for HER yet, and are inferior to other nickel based compounds [9]. By modifying the technique of H Schäfer et al., we were able to create a nanostructured Ni-Fe surface on the SSM electrode. These nanostructures work as a highly-efficient bi-functional catalyst for the full water splitting reaction and the final OER catalyst is synthesized in only 5 minutes and consist of Ni-Fe nanosheets with a structure similar to Ni-Fe LDH catalysts found in the literature [5,14,15,16]. Our final electrodes need 1.78 V to produce a current density of 10 mA × cm^−2^ with an onset potential of only 1.68 V for the full water splitting reaction. This result is barely 0.1 V higher compared to the most efficient nanosized bi-functional electrocatalysts adhered on high surface electrodes and comparable to nanosized Ni-Fe based catalysts [14,17,18,19,20]. We believe that this top-down approach increases the electrical and physical connection between the catalyst and current collector by removing the interface between them, and can be a cost-efficient alternative compared to standard electrocatalyst-coated electrodes.

## 2. Materials and Methods

Material synthesis: As starting material stainless steel (type AISI 316L) mesh (SSM-316L) with mesh size 500 and wire diameter of ~25 µm was used. The SSM was etched by sonication in 2M HCl for 45 minutes. At the end of the etching process the temperature reached ~45 °C and thereafter the SSM was cleaned by further sonication in ethanol and Milli-Q water (Milli-Q-Plus, Merck Millipore, Burlington, MA, USA) for 15 min each. The anodization of a SSM piece (1 cm^2^) was performed in a two-electrode setup in 7.5 M NaOH held at 50 °C with an applied potential of 5.0 V for 5 min with a large sheet of SSM as counter electrode. The current reached 3 A and was constant during the anodization. The anodization was also performed by varying the potential from 3.5 V to 6.5 V and between 5 min up to 4 h without any further improvement of the performance. Finally, the sample was sonicated in ethanol and Milli-Q water and dried at 50 °C. The sample was further activated for HER by reduction in a closed furnace with a constant flow 180 ml/min of Varigon (Ar:H_2_ 95:5, AGA Gas, Sundsvall, Sweden) during the whole procedure. First, the temperature was increased to 250 °C and held for 30 minutes and then further increased to 450 °C and held for 1 hour. The temperature of the furnace was then naturally decreased to room temperature for ~5 h before opening the furnace.

Characterizations: X-ray photoelectron spectroscopy (XPS) was performed on a Kratos axis ultra-delay line detector electron spectrometer using a monochromatic Al-K_α_ source operated at 150 W. The analysis area of the spectrometer was 0.3 mm × 0.7 mm. Binding energy scale of the spectrometer was calibrated using ISO standard 15,472:2010 SCA-XPS-Calibration of energy scale [21]. Accuracy in the BE position determination is 0.1 eV and in atomic ratios calculations better than 10%.

Scanning electron microscopy (SEM) was performed on a Zeiss Merlin FESEM (Carl Zeiss, Oberkochen, Germany) at 5 kV and 120 pA and EDS data was measured at 15 kV and 400 pA. Raman spectroscopy was conducted on a Renishaw InVia Raman spectrometer with a charge-coupled device detector. As excitation source a laser with wavelength of 514 nm was used. XRD was done with a PANalytical X’pert3 powder X-Ray diffractometer.

Electrochemical measurements were performed in a three-electrode setup connected to a potentiostat (Metrohm-Autolab PGSTAT302N, Herisau, Switzerland). The working electrode consisted of a SSM piece attached to a glass substrate with double sided carbon tape and the active area was fixed to 0.2 cm^2^ by shielding the surroundings with epoxy resin. The electrode was then connected to the instrument via a copper wire. The reference electrode was an Ag/AgCl electrode (CHl111-CH instruments, Austin, TX, USA) stored in 1M KCl and the potentials were referred to RHE by adding 0.222 + 0.059*pH. The counter electrode was a coiled platinum wire (99.99%) in all measurements. For the linear sweep voltammetry the potential sweep rate was 10 mV/s with a potential step of 5 mV. Cyclic voltammetry was conducted at 50 m/Vs. For determining the full cell performance, two-terminal measurements are made with the electrodes separated by 1 cm from each other. In all electrochemical measurement, 1.0 M KOH is used as the electrolyte and no IR-corrections were done.

## 3. Results

### 3.1. Material Synthesis and Characterization

Initially, we start with a commercial stainless steel mesh (SSM-316L), which has a plane and chromium rich surface (Appendix A) that are poor features for an electrocatalyst. In order to activate the SSM for the water splitting reaction, we use a top-down approach to modify the intrinsic characteristics of the SSM. Figure 1 shows a schematic of the activation procedure (details can be found in the experimental section). Firstly, to increase the surface area, we etched the SSM by sonication in HCl for 45 min. During the experiment, the temperature increased to 45 °C. The sonication in acid results in a SSM with a nano-sized porous structure (Appendix A) and a concurrent increase in the amount of nickel atoms at the surface compared to the initial SSM-316L (Appendix A). The etching process was optimized by changing HCl concentration (1M–4M), etching time (30 min–3 h), and temperature (20 °C–70 °C). Accordingly, we found that 1M HCl and 45 min were optimal, and at temperatures higher than 50 °C nickel dissolved, turning the solution green and deformed the SSM. Continuing with the acid-etched SSM, we further anodized it in a highly alkaline electrolyte (7.5M NaOH), modifying the technique of Schäfer at al. [12] slightly (See experimental section for details). After anodizing the SSM for only 5 minutes, we noticed a thin film of nanosheets form on the surface and this sample is hereafter labeled as SSM-A. The nanosheets are shown in the SEM micrographs in Figure 2a and Appendix A with a morphology similar to layered double hydroxides (LDH) found in the literature [3,5].

While highly oxidized electrocatalysts are known to perform well for the OER, reduced metal oxides or metal chalcogenides are normally used as catalysts for HER [2,4,6,22]. Thus, to further optimize the surface characteristics for HER, we annealed SSM-A in a reducing atmosphere at 400 °C for 1 hour that fused the nanosheets into nanoparticles (10–100 nm) shown in Figure 2b and Appendix A. This sample is hereafter labeled as SSM-AR.

It is widely accepted that catalytic activities are controlled by elements located near or at the outermost surface of catalysts, by altering the adsorption energies of reaction intermediates [23,24]. It is therefore important to study the elemental composition and chemical states at the surface of an electrocatalyst with surface sensitive techniques such as X-ray photoelectron spectroscopy (XPS). Hence, we conducted XPS of the initial SSM-316L, the same sample but after acid-etching and for the final catalysts, SSM-A and SSM-AR. In all XPS data (Figure 3, Appendix A), we only present the binging energy for the main peak of each chemical state and not for any doublet or satellite peaks. As mentioned, chromium oxide is electrically insulating and could impair the catalytic activity, contrary to transition metals like nickel, iron, and molybdenum, which has found to have optimal properties for OER and HER catalysis [3,5,17,25,26,27,28]. In Table 1 we show the metal concentrations on the surfaces and for the initial SSM, iron and chromium oxides were the predominant metals found, while the nickel signal was even below the detection limit in our setup. As suspected, this is not an optimal surface for an electrocatalyst, which can be seen by the low electrochemical activities in Figure 4 for SSM-316L, both for HER and OER. On the acid-etched SSM however, nickel becomes detectable (see Appendix A), but only at low atomic concentrations, while is chromium still the predominant metal based on the XPS analysis.

For the anodized SSM-A, as shown in Table 1, the surface changes entirely. While the chromium signal disappears, nickel is evident at the surface after the anodization. This is in line with a similar study and is believed to be a result of leached Cr and that nickel migrates to the surface under harsh anodic conditions [12]. We also noted that particles were found on the counter electrode after anodization with a Ni:Fe:Cr ratio of 0.07:1.00:0.06. After normalizing the integrated intensities to total metal intensity (and removing C1s and O1s signals) SSM-A showed nickel and iron atomic concentrations of 65% and 35%, similar to optimal ratios found in recent studies [15,29,30]. Interestingly, this happens already after 5 minutes of anodization at 50 °C.

Figure 3a compares the high-resolution Ni2p spectral lines between SSM-A and SSM-AR (no signal was detected on the initial SSM). It is clear the SSM-A only consist of highly oxidized chemical states, such as Ni(II)(OH)_2_ or Ni(III)OOH, which are difficult to separate from each other with XPS [31]. However, it is known that nickel oxidizes to NiOOH during OER in alkaline solution, which is the active chemical state for OER and could explain that this is the predominant chemical state for nickel in SSM-A [16,30]. A highly oxidized chemical state is also evident in the Fe2p spectral line of SSM-A and is assigned to Fe(III)OOH [12,31]. In Table 1, the elemental composition as well as metal-hydroxides (M-OH) to metal-oxides (M-O) ratio of all samples is shown. The SSM-A is highly oxidized and displays a M(OH):M(O) ratio of 3.28. In Appendix A, detailed XPS data are shown for all samples. Raman spectroscopy of the SSM-A displayed in Appendix A, confirms Ni-Fe layered double hydroxides (LDH) and thus supports the XPS data [32].

Taking the XPS, SEM and Raman results into account, we believe the nanosheets at the surface of SSM-A to be Ni-Fe LDH, similar to nanostructures found in literature [3,5].

Continuing with the annealed SSM-AR, the XPS spectra show that the metals are slightly reduced with a M-OH:M-O ratio of only 0.50. Some of the Ni and Fe metals in SSM-AR are even fully reduced to pure metal states, which were detectable with XPS (Figure 3, Appendix A) [31]. Regarding the Ni:Fe ratio, it also decreases to 0.6:1.0, thus displaying an increased surface concentration of Fe in SSM-AR compared to SSM-A. The Raman spectrum in Appendix A also confirms that the surface chemical states have been reduced and found mostly as Fe(II/III) oxides [33]. Additionally, in SSM-AR, some chromium was detectable after the annealing, which could have an inferior effect on the electrical conductivity. While molybdenum is preferred in HER catalysts and present in the SSM bulk, it could not be detected in SSM-AR, probably due to the low concentration in the initial SSM and the absence at the surface of SSM-A. In summary, we believe that the Ni-Fe metals are still responsible of the electrocatalytic HER activity but at reduced chemical states, such as Ni(II)(O) and Fe(II/III)(O).

Finally, we conducted XRD on the samples for crystallographic information. Appendix A shows the XRD diffractograms, which only display weak diffractions corresponding to austenitic steel from the bulk structure in all samples [34]. The reason for an absent signal of surface crystals is probably because of the structure of the SSM fibers, and the fact that the nanosheets and nanoparticles only forms as thin films on the surface and are therefore not detectable by XRD.

### 3.2. Electrochemical Performance

The electrochemical performance was evaluated for the electrochemically activated SSM-A and SSM-AR in a three electrode cell with Ag/AgCl and Pt as reference and counter electrodes. In Figure 4, polarization curves of SSM-A from linear sweep voltammetry shows that the catalytic activity for OER drastically increases compared to the initial SSM-316L and that it even outperforms commercial IrO_2_ (2 mg/cm^2^) adhered on carbon paper. For OER, SSM-A needs only 1.51 V for producing a current density of 10 mA/cm^2^ and displays an onset potential as low as 1.49 V, which is comparable to high performing nanosized electrocatalysts for OER [4]. This result is similar with previous studies of activated stainless steel [9,12]. The oxidation peak seen at 1.49 V corresponds to oxidation of nickel species. Considering HER, the catalytic activity for SSM-AR drastically improves after activation, and outperforms both SSM-A and the initial SSM-316L. For HER, SSM-AR need 0.28 V to reach a current density of 10 mA/cm^2^ with an onset potential of 0.25 V. This performance is excellent compared to the initial SSM and SSM-A, which need 140 mV and 190 mV higher onset potentials for HER, respectively. However, the activity for HER is still considerably lower than state of the art catalyst. In agreement with previous studies, we believe that removing the low concentrations of Cr and introduce more molybdenum at the surface, could even boost the catalytic activity further for HER, thus lowering the total overpotential [17,25,28]. Furthermore, the activity can be increased further by creating a porous structure such as in Raney-Nickel structures [9]. Consequently, if the potential could be decreased by 0.1 V for HER, the SSM would be comparable to the best performing electrocatalysts for the full water splitting reaction [4].

Long-term performance stability tests were done on SSM-A and SSM-AR using cyclic voltammetry. Appendix A displays a high cyclic stability for both catalysts were only 10 mV and 30 mV higher potentials were needed for SSM-A and SSM-AR to achieve current densities of 10 mA/cm^2^ after 1000 sweeps compared to the initial sweep. Furthermore, the performance for the full water splitting reaction was tested by connecting the two electrodes 1 cm apart in a head-to-head two electrode-setup configuration. In Figure 5 the polarization curve for the full water splitting reaction is shown, and in 1.0M KOH the onset potential was only 1.68 V while 1.78 V was needed for a current density of 10 mA/cm^2^. This performance is comparable to other highly efficient bi-functional nano-sized electrocatalyst adhered on high surface area 3D electrodes, and only slightly below the most efficient nano-sized bi-functional electrocatalysts so far [18,26,35].

## 4. Discussion

We successfully activated plain stainless steel mesh (SSM, AISI 316L) for catalyzing the full water splitting reaction using a top-down approach where we firstly anodized the SSM under harsh conditions to work as anode (SSM-A) and subsequently reduced it in high temperature to work as cathode (SSM-AR). An onset potential of 1.68 V and a current density of 10 mA/cm^2^ at 1.78 V where achieved when using the SSM-A and SSM-AR in a two-terminal electrochemical cell, which is significantly improved compared to the initial SSM-316L. We address the increased activity to a superior surface area and to the migration of Ni and Fe to the surface leading to the formation of Ni-Fe nanostructures at the surface. Nickel and iron layered hydroxides are believed to be the active chemical states for OER and slightly reduced iron and nickel oxides nanoparticles for HER. Considering OER, the activity of Ni-Fe layered nanosheets in SSM-A is even comparable to the current best-performing nanocatalysts with an onset potential of only 260 mV. However, we believe that the HER could be further improved if the bulk molybdenum could be utilized further or that the sizes of nanoparticles are more controllable. To achieve this, a starting material with a higher initial atomic concentration of molybdenum than stainless steel (316L) could favorably be used. Still, by using a facile top-down approach, we show that stainless steel has potential to be used as the electrocatalyst itself and not only as current collector in alkaline electrolyzers. Finally, we believe using a norm-breaking top-down approach when preparing electrodes guarantees an excellent electrical and physical contact between the electrocatalyst and the current collector, which could be beneficial when constructing commercial electrolyzers.

## Figures and Tables

**Figure 1 materials-12-02128-f001:**
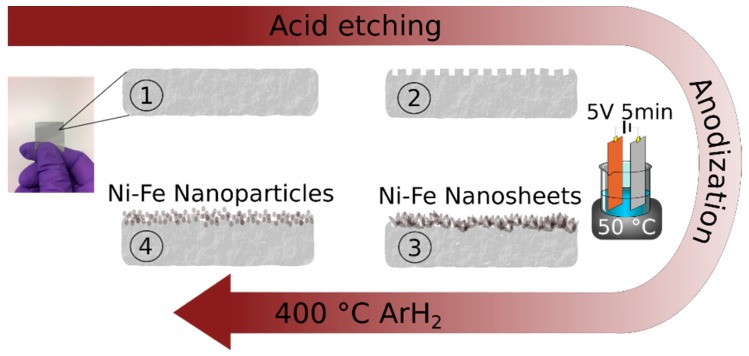
A schematic showing the activation procedure. Firstly, in ① a stainless steel mesh (SSM-316L) was cleaned and degreased. In ② the SSM was sonicated in in an acidic solution to increase the surface area. In ③ the SMM was anodized in a 7.5 M NaOH solution by applying 5 V for 5 min to generate nickel-iron nanosheets (SSM-A). Finally in ④ the sample is further annealed in a reducing atmosphere to form nanoparticles (SSM-AR).

**Figure 2 materials-12-02128-f002:**
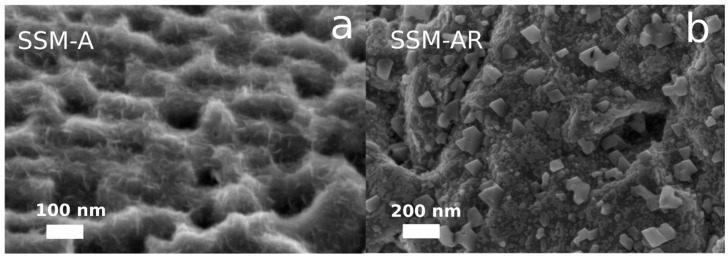
SEM micrographs of (**a**) the anodized SSM (SSM-A) displaying Ni-Fe nanosheets, and (**b**) the reduced SSM (SSM-AR) where the nanosheets agglomerate into nanoparticles. Note the different scalebars.

**Figure 3 materials-12-02128-f003:**
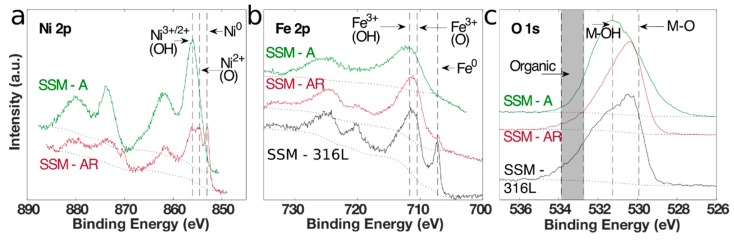
X-ray photoelectron spectroscopy spectra of initial SSM-316L, SSM-A and SSM-AR showing the (**a**) Ni2p, (**b**) Fe2p, and (**c**) O1s spectral lines. The labels shows the oxidation state with superscript and additionally if the metal-oxygen peaks involves oxygen (O) or hydroxide (OH). All peak positions are taken from the deconvoluted spectra.

**Figure 4 materials-12-02128-f004:**
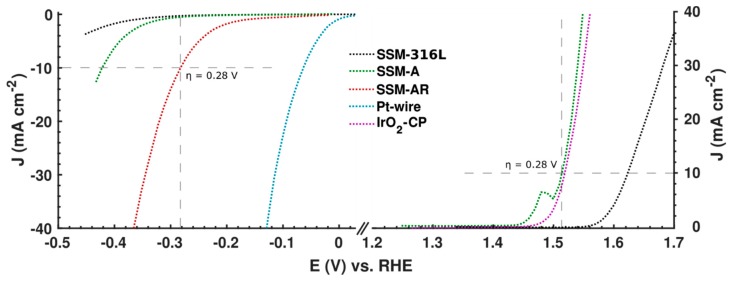
Polarization curves for the hydrogen evolution reaction (left) and oxygen evolution reaction (right) comparing the SSM electrocatalysts along with Pt and IrO_2_ reference samples. Electrochemical data were recorded with linear sweep voltammetry in 1.0 M KOH electrolytes with a scan rate of 5 m/Vs. The reference and counter electrodes used were an Ag/AgCl (1M KCl) electrode and a Pt-coil.

**Figure 5 materials-12-02128-f005:**
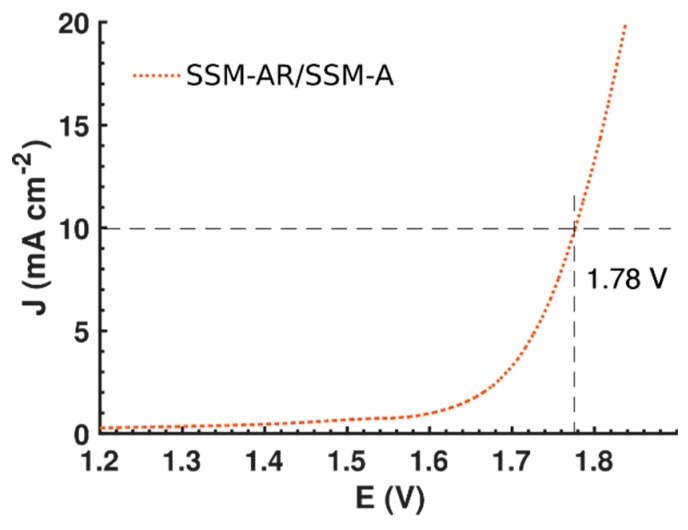
A LSV showing the performance for the full water splitting reaction, using SSM-A as the OER catalyst and SSM-AR as the HER catalyst. At a potential difference of 1.78 V (or an overpotential of 0.55 V) the two electrodes produce a current density of 10 mA/cm^2^. The scan rate for the LSV was 2 mV/s and it was conducted in 1.0 M KOH. The separation distance between the electrodes was 1 cm.

**Table 1 materials-12-02128-t001:** Atomic percentages, normalized to total metal concentration obtained with XPS. The M-OH:M-O ratio is taken as the integrated intensity ratio between the two metal-oxides peaks in the O 1s spectral line with BE defined at 530.0 eV (M-O) and 531.5 eV (M-OH).

Sample	Ni (at. %)	Fe (at. %)	Cr (at. %)	Mo (at. %)	M-OH:M-O Ratio
Initial SSM-316L	-	67	32	1.1	0.85
SSM-A	65	35	-	-	3.28
SSM-AR	32	54	14	-	0.50

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
