# Peer review of "Stainless Steel as A Bi-Functional Electrocatalyst—A Top-Down Approach"

_materials, 2019, doi:10.3390/ma12132128_

Round 1

Reviewer 1 Report

The paper presents results of stainless steel as a bi-functional electro-catalyst. The paper contains interesting results, which can be in principle published in Materials.

The manuscript is well written. Moreover, the Figures are well formatted. However, the authors must explain the significance of their work to be published in Materials from their point of view. It seems that there are some papers using stainless steel as catalyst for HER and OER.

I see some potential in the present manuscript. However, I have the following comments, which must be properly addressed, before the paper can be accepted for publication:

Page 1. Abstract. The authors should include alkaline or acidic media, range of temperature, concentration and the current density for 1.68 V of onset potential.

Page 1. Introduction. The authors should point out the medium (alkaline) before starting the discussion about catalyst.

Page 1. Line 44. What about Raney-Nickel?

Page 2. Lines 51-53. There are others papers studying stainless steel for HER and OER in alkaline (https://www.mdpi.com/1996-1944/12/8/1336). The authors should discuss it at the introduction.

Page 2. Lines 58-59. Remove the sentence. stainless steel has not been used as an efficient electro-catalysis due to Raney-Niquel is quite better.

Page 3. Line 109. I can not see the supporting information.

Page 3. Lines 115-116. I can not see figure S1 and Table S1.

Page 4. Figure 2. The legend should include a) and b).

Page 4. Figure 3. The legend should include a), b) and c).

Page 5. 3.2 Electrochemical performance. I think that a double scale overpotential and E vs RHE would be more usefull for figure 4 a) and b). Also, for HER Authors must compare their results with Raney-Nickel for Figure 4 a) and with stainless steel 316 and 304 in Figure 4 b). In the previous paper (https://www.mdpi.com/1996-1944/12/8/1336) the authors can find some references.

Page 6. Figure 1. Should be Figure 4...The legend should include a) and b).

Page 6. Figure 2. Should be Figure 5...What about the cell resistance?. Also, a graphical comparison with other authors would be desirable.

Finally, I think if Authors find some of the mentioned references useful, they should include them in the reference list section of the present manuscript. With the aim of showing to the reader other points of view or confirmation of the present hypotheses, which will add impact to the present proposal.

I would appreciate receiving the revised version of this manuscript!

Author Response

Dear Editor,

We thanks the reviewer and the editor for the quick and professional handling of our manuscript. We have addressed all points raised by the reviewers and below we give a point-by point response on the various issues. Please do not hesitate to contact us in case you have any inquiries,

Best Regards

Thomas Wågberg, Professor in Physics,

Review 1:

Open Review

English language and style

( ) Extensive editing of English language and style required 
( ) Moderate English changes required 
( ) English language and style are fine/minor spell check required 
(x) I don't feel qualified to judge about the English language and style 

Yes

Can be improved

Must be improved

Not applicable

Does the introduction   provide sufficient background and include all relevant references?

( )

( )

(x)

( )

Is the research design   appropriate?

( )

(x)

( )

( )

Are the methods   adequately described?

(x)

( )

( )

( )

Are the results   clearly presented?

(x)

( )

( )

( )

Are the conclusions   supported by the results?

( )

(x)

( )

( )

Comments and Suggestions for Authors

The paper presents results of stainless steel as a bi-functional electro-catalyst. The paper contains interesting results, which can be in principle published in Materials.

The manuscript is well written. Moreover, the Figures are well formatted. However, the authors must explain the significance of their work to be published in Materials from their point of view. It seems that there are some papers using stainless steel as catalyst for HER and OER.

We thank the reviewer and tried to highlight in the introduction that we create and investigate nanostructures on the stainless steel, which we have never seen in other papers about stainless steel electrodes.

I see some potential in the present manuscript. However, I have the following comments, which must be properly addressed, before the paper can be accepted for publication:

Page 1. Abstract. The authors should include alkaline or acidic media, range of temperature, concentration and the current density for 1.68 V of onset potential.

Response: We have added “At ambient temperature, the activated stainless steel electrodes produces 10 mA cm-2 at a cell voltage of 1.78 V and an displays an onset for water splitting at 1.68 V in 1M KOH” in the abstract.

Page 1. Introduction. The authors should point out the medium (alkaline) before starting the discussion about catalyst.

Response: Agree: we have clarified when we discuss about alkaline or acidic in the introduction.

Page 1. Line 44. What about Raney-Nickel?

Response: We thank the reviewer for giving this suggestion. We have added nickel as a catalyst in the reference list and cited the paper https://www.mdpi.com/1996-1944/12/8/1336.

Page 2. Lines 51-53. There are others papers studying stainless steel for HER and OER in alkaline (https://www.mdpi.com/1996-1944/12/8/1336). The authors should discuss it at the introduction.

Response: We added following paragraph in the introduction: We must mention that stainless steel recently have shown to be active as an electrocatalyst and in this study we focus to create a nanostructured surface to even improve the high activities measured. We referred to three papers here, among one mentioned in https://www.mdpi.com/1996-1944/12/8/1336.

Page 2. Lines 58-59. Remove the sentence. stainless steel has not been used as an efficient electro-catalysis due to Raney-Niquel is quite better.

Response: We have modified this sentence to: To our knowledge, stainless steel has not showed high efficiency as an electrocatalyst for HER yet, and are inferior to other nickel based compounds.

Page 3. Line 109. I can not see the supporting information.

Response: It seems as if a problem occurred when uploading the supporting information. We hope that this problem is solved for this submission.

Page 3. Lines 115-116. I can not see figure S1 and Table S1.

Page 4. Figure 2. The legend should include a) and b).

Response: We agree with the reviewer and have added a and b in Figure 2.

Page 4. Figure 3. The legend should include a), b) and c).

Response: We have also added a,b,c in the legend here. The new figure is inserted in the manuscript.

Page 5. 3.2 Electrochemical performance. I think that a double scale overpotential and E vs RHE would be more usefull for figure 4 a) and b). Also, for HER Authors must compare their results with Raney-Nickel for Figure 4 a) and with stainless steel 316 and 304 in Figure 4 b). In the previous paper (https://www.mdpi.com/1996-1944/12/8/1336) the authors can find some references.

Response: We added for OER: This result is similar with previous studies of activated stainless steel.

We added for HER: However, the activity for HER is still considerably lower than state of the art catalyst.

Furthermore, creating a porous structure such as in Raney-Nickel should increase the activity further

Page 6. Figure 1. Should be Figure 4...The legend should include a) and b).

Response: We have added the a and b in Figure 4 to one combined figure. It seemed to be a formatting error in the generated pdf file in a way that the numbering of the figures were wrong. This is not present in the word-file and the pdf generation upon submission is out of our control.

Page 6. Figure 2. Should be Figure 5...What about the cell resistance?. Also, a graphical comparison with other authors would be desirable.

Response: We have not corrected for the cell resistance but since the electrolyte is 1M KOH and the separation distance is only 1 cm the resistance should be around 2 ohm and at such low current densities it would not affect the appearance significantly. However, due to lack of instrumentation in this setup we have not measured the cell resistance for this particular cell.

Finally, I think if Authors find some of the mentioned references useful, they should include them in the reference list section of the present manuscript. With the aim of showing to the reader other points of view or confirmation of the present hypotheses, which will add impact to the present proposal.

Response: We have added some of these references.

I would appreciate receiving the revised version of this manuscript!

Review 2

Open Review

English language and style

( ) Extensive editing of English language and style required 
( ) Moderate English changes required 
(x) English language and style are fine/minor spell check required 
( ) I don't feel qualified to judge about the English language and style 

Yes

Can be improved

Must be improved

Not applicable

Does the introduction   provide sufficient background and include all relevant references?

(x)

( )

( )

( )

Is the research design   appropriate?

(x)

( )

( )

( )

Are the methods   adequately described?

(x)

( )

( )

( )

Are the results   clearly presented?

( )

(x)

( )

( )

Are the conclusions supported   by the results?

( )

(x)

( )

( )

Comments and Suggestions for Authors

This paper expands on the use of stainless steel as precursor for active OER and HER electrocatalysts. It is an interesting work showing that by using pre-treatment methods, surface modification occurs leading to the formation of highly active electrocatalysts for water splitting.

The authors have explained properly the methodology and the new materials have been characterised fully with techniques such as XPS and SEM. I couldn't have access to the SI which limited slightly my ability to judge some of the comments made in the paper, as well as to see the results from Raman spectroscopy. The electrochemical analysis is basic and shows clearly the increase in performance on the catalysts. A few points need to be tackled before publication is accepted:

- Some minor typos in the language with plurals and singulars not correctly used.

Response: We thank the reviewer for finding this and we have read through the text carefully and tried to find all of these small errors.

- Figure numbering is incorrect.

Response: We have noticed this and it seemed to be a formatting error in the generated .pdf file that the numbering of the figures were wrong.

- In the introduction, it should be stressed that this paper deals with alkaline electrolysis, in particular when talking about the materials used.

Response: We have clarified this in the introduction.

- The formation of SSM-AR under reducing atmosphere, can this be performed under cathodic treatment? 

Response: This is a good suggestion and we actually tried that. Unsuccessfully we were not able to increase the activity or create similar structure with this method.

- Figure 2 should show both images at the same scale.

Response: We have now added a scalebar with similar size that shows 200 nm in b. Unfortunately we did not have high resolution images with the same scale of this batch. We believe that the nanostructures are well displayed in this figure though, and in supplementary Figure S1 we have all 4 samples with the same magnification.

- Figure 3: there are peaks that have not been assigned in all spectra. Have the authors shown in SI how the peaks have been deconvoluted to quantify metal quantities?

Response: Thanks for stating this, we have actually assigned all main peaks in these spectra. We clarified with the text: In all XPS data (Figure 3, Table S1), we only present the binging energy for the main peak of each chemical state and not for any doublet or satellite peaks. In Table S1 all the fitted peak positions are listed. The rest of the peaks shown in the spectra are doublet peaks and satellites for Ni and Fe. We hope that we clarified this enough in the reviewed version.

- Have the authors checked using ICP the composition of the electrolyte after anodization? Is it metal migration of metal leaching to the solution the mechanism to activate the electrode surface?

Response: Unfortunately we do not have access to ICP so we have not done these measurements. However, a previous similar study did this and found out that nickel migrated to the surface while Fe and Cr was leached in the solution during anodizxation. We did XPS on the CE to investigate electrodeposited elements. We have added the text: This is in line with a similar study and is believed to be a result of leached Cr and that nickel migrates to the surface under harsh anodic conditions.12 We also noted that particles were found on the counter electrode after anodization with a Ni:Fe:Cr ratio of 0.07:1.00:0.06.

- Authors should show CV or LSV after 1000 cycles for comparison.

Response: The CV are shown in supplementary figure S5 for both HER and OER.

- Figure 4 for SSM-A, the authors should identify the peak at 1.48V.

Response: We agree with this and have added in the text: The oxidation peak seen at 1.49 V corresponds to oxidation of nickel species.

Review 3

Open Review

English language and style

( ) Extensive editing of English language and style required 
( ) Moderate English changes required 
(x) English language and style are fine/minor spell check required 
( ) I don't feel qualified to judge about the English language and style 

Yes

Can be improved

Must be improved

Not applicable

Does the introduction   provide sufficient background and include all relevant references?

(x)

( )

( )

( )

Is the research design   appropriate?

(x)

( )

( )

( )

Are the methods   adequately described?

(x)

( )

( )

( )

Are the results   clearly presented?

(x)

( )

( )

( )

Are the conclusions   supported by the results?

(x)

( )

( )

( )

Comments and Suggestions for Authors

Stainless steel mesh was chemically and electrochemically activated and checked for its water splitting capabilities. The results with respect to electrochemical activity and stability are convincing to me. The manuscript is well written and should be accepted for publication in Materials MDPI.

I found only one point that should be corrected prior to publication (see below)

Abstract line 17-19:

 “The activated stainless steel electrodes displays an onset potential of 1.68 V for the full water splitting 18 reaction, which is close to benchmarking nanosized catalysts”

This is not a potential but a potential difference. My suggestion: “… displays an onset of water splitting at a cell voltage of 1.68 V.

Response: We thank the reviewer for the suggestion and have changed the text accordingly: At ambient temperature, the activated stainless steel electrodes produces 10 mA cm-2 at a cell voltage of 1.78 V and an displays an onset for water splitting at 1.68 V in 1M KOH

Reviewer 2 Report

This paper expands on the use of stainless steel as precursor for active OER and HER electrocatalysts. It is an interesting work showing that by using pre-treatment methods, surface modification occurs leading to the formation of highly active electrocatalysts for water splitting.

The authors have explained properly the methodology and the new materials have been characterised fully with techniques such as XPS and SEM. I couldn't have access to the SI which limited slightly my ability to judge some of the comments made in the paper, as well as to see the results from Raman spectroscopy. The electrochemical analysis is basic and shows clearly the increase in performance on the catalysts. A few points need to be tackled before publication is accepted:

- Some minor typos in the language with plurals and singulars not correctly used.

- Figure numbering is incorrect.

- In the introduction, it should be stressed that this paper deals with alkaline electrolysis, in particular when talking about the materials used.

- The formation of SSM-AR under reducing atmosphere, can this be performed under cathodic treatment? 

- Figure 2 should show both images at the same scale.

- Figure 3: there are peaks that have not been assigned in all spectra. Have the authors shown in SI how the peaks have been deconvoluted to quantify metal quantities?

- Have the authors checked using ICP the composition of the electrolyte after anodization? Is it metal migration of metal leaching to the solution the mechanism to activate the electrode surface?

- Authors should show CV or LSV after 1000 cycles for comparison.

- Figure 4 for SSM-A, the authors should identify the peak at 1.48V.

Author Response

(The authors gave the same response as above.)

Reviewer 3 Report

Stainless steel mesh was chemically and electrochemically activated and checked for its water splitting capabilities. The results with respect to electrochemical activity and stability are convincing to me. The manuscript is well written and should be accepted for publication in Materials MDPI.

I found only one point that should be corrected prior to publication (see below)

Abstract line 17-19:

 “The activated stainless steel electrodes displays an onset potential of 1.68 V for the full water splitting 18 reaction, which is close to benchmarking nanosized catalysts”

This is not a potential but a potential difference. My suggestion: “… displays an onset of water splitting at a cell voltage of 1.68 V.

Author Response

(The authors gave the same response as above.)

Round 2

Reviewer 1 Report

Authors have improved remarkably the manuscript. I think it is ready to be published in the present form.

Author Response

The reviewer did not ask for any further changes. We have made some editorial changes following the remarks from the editor,

Best Regards

Thomas